# Basic Research for Additive Manufacturing of Rubber

**DOI:** 10.3390/polym12102266

**Published:** 2020-10-01

**Authors:** Welf-Guntram Drossel, Jörn Ihlemann, Ralf Landgraf, Erik Oelsch, Marek Schmidt

**Affiliations:** 1Professorship for Adaptronics and Lightweight Design in Production, Faculty of Mechanical Engineering Chemnitz University of Technology, 09126 Chemnitz, Germany; adaptronik@mb.tu-chemnitz.de; 2Fraunhofer Institute for Machine Tools and Forming Technology IWU, 09126 Chemnitz, Germany; 3Chair of Solid Mechanics, Faculty of Mechanical Engineering, Chemnitz University of Technology, 09126 Chemnitz, Germany; joern.ihlemann@mb.tu-chemnitz.de (J.I.); ralf.landgraf@mb.tu-chemnitz.de (R.L.); erik.oelsch@mb.tu-chemnitz.de (E.O.)

**Keywords:** 3D rubber printing, additive manufacturing, screw extruder

## Abstract

The dissemination and use of additive processes are growing rapidly. Nevertheless, for the material class of elastomers made of vulcanizable rubber, there is still no technical solution for producing them using 3D printing. Therefore, this paper deals with the basic investigations to develop an approach for rubber printing. For this purpose, a fused deposition modeling (FDM) 3D printer is modified with a screw extruder. Tests are carried out to identify the optimal printing parameters. Afterwards, test prints are performed for the deposition of rubber strands on top of each other and for the fabrication of simple two-dimensional geometries. The material behavior during printing, the printing quality as well as occurrences of deviations in the geometries are evaluated. The results show that the realization of 3D rubber printing is possible. However, there is still a need for research to stabilize the layers during the printing process. Additionally, further studies are necessary to determine the optimum parameters for traverse speed and material discharge, especially on contours.

## 1. Introduction

At present, there is a wide range of materials that can be processed using various additive methods and the market continues to grow rapidly. Metallic powder materials can be melted into extremely fine filigree structures using laser and beam-based processes [1,2]. Furthermore, there are additive processes for additional material classes such as polymers and ceramics. Worth mentioning here is selective laser sintering (SLS). In this additive process, a thin powder layer of the material is melted in local areas, which forms a solid material layer after solidification. During this process, the melting point of the material is not exceeded. Subsequently, another layer of powder is applied and melted in the local areas. In this way, a component is created layer by layer. After completion, the excess powder will be removed. Processable materials include metals and ceramics [3,4], but polymers can also be manufactured [5].

Fused deposition modeling (FDM) or also called fused filament fabrication (FFF) is another additive process for polymeric materials. In this process, materials such as meltable plastics or thermoplastic elastomers (TPE) are built up layer-by-layer to form a component [6]. The 3D printing process used for the longest time is stereolithography (SLA). The material used in this case is a photopolymer, which is initially available as a liquid bath. A laser scans the desired component shape and hardens the material. The hardened layer is then lowered and the next layer is produced [7]. Polyjet or multijet modeling (PJM/MJM) works on a different principle. Liquid acrylic polymers are applied drop-by-drop onto a building platform via a print head with one or more nozzles. The material is then cured layer-by-layer by irradiation with UV light [8]. Furthermore, there are less common additive processes such as binding jetting [2], and layer laminated manufacturing [9], which also cover the material classes of polymers, metals and ceramics.

Although there are numerous additive processes as well as usable materials, there is still no technical solution for the additive manufacturing of elastomers made of vulcanizable rubber. Due to their specific mechanical properties, elastomers have become indispensable in many everyday areas. Compared to other polymers, properties such as high mechanical strength, very high elongation at break and good impact resilience, to name but a few, distinguish the elastomers. As with other classes of materials, the possibility of manufacturing components using additive processes in areas of the elastomer industries, would bring decisive advantages, such as freedom in the designs. Up to now, the production of elastomer molded parts is dominated by automated injection molding. Further options are, among others, compression molding and thermoforming. Further details of the above procedures can be found in [10,11,12]. However, what all the procedures have in common is the requirement for previous production of expensive molding tools, which also severely limits the individuality and subsequent customization. Elastomer production using 3D printing would enable manufacturing of the first component samples at an early stage in the product development. Optimizations and individual changes can be implemented faster and development loops would be shortened accordingly [13].

Research work in this area is worthwhile since there is a need for the additive manufacturing of elastic materials. This is shown, not least, by the increasing use of additive processing of the above-mentioned TPE using the FDM process. In addition to other differences from elastomers, TPE has considerable disadvantages in terms of temperature stability. Due to melting at elevated temperatures, the service temperature is well below the melting point. Elastomers made of rubber can usually be exposed to these temperatures for a short time [14]. Furthermore, there is a technological solution for UV-curing silicon elastomers from ACEO, a brand of Wacker Chemie AG. The specially developed “drop on demand process” works in a manner similar to the PJM/MJM processes [15]. Digital light processing can also be used for silicon elastomers [16], although, there is always the risk of embrittlement by ambient light during the use of UV-curing materials. A limited realization for the additive manufacturing of vulcanizable rubbers is provided by [17]. The formulation of two rubber components is designed in such a way that self-vulcanization occurs when they are combined. In 3D printing, the material is merged in the extruder so that it has sufficient viscosity until it leaves the nozzle. Subsequently, vulcanization begins on the building platform, which keeps the geometrically desired shape stable. However, the possible properties of the final part remain very limited due to the mainly specified chemical composition of the rubber compounds.

A major reason why there is still no additive technology for elastomers is the complex material composition of vulcanizable rubbers. Due to the resulting rheological processing behavior in conjunction with the required chemical cross-linking, the reaction requires a complex process chain. In contrast to thermoplastic materials, vulcanizable rubber will not harden after being discharged from the nozzle. Above a critical layer height, the structure deforms due to its own weight, so that an accurate build-up process based on the shape is impossible. In addition, the subsequent vulcanization and crosslinking of the rubber causes a reduction in the viscosity of the rubber compound due to the increase in temperature, which results in a melting of the built-up structure and a total loss of the geometric shape. Therefore, the rheological properties and the requirement of vulcanization for transferring into an elastomer result in the following significant challenges for the additive process:a suitable method and process technology for processing and discharging the rubber material,supporting materials during the layer build-up, as well asmaterial for stabilizing the shape during vulcanization.

The aim of this study is to investigate the extrusion-based process for 3D-printing of rubber, in order to test the first mentioned challenge in technological terms. In contrast to the FDM process, where the feed material has to be in filament form, the use of a screw extruder offers more possibilities regarding the form of the feed material. The material can be processed, for example, in the form of powder, granulates, or liquid or rubber stripes. In particular, extrusion-based 3D printing is already used for non-thermoplastic materials such as ceramics [18], or cement-based composites [19]. Often biomaterials are also processed using this additive procedure [20]. Furthermore, the extrusion-based process is also a research area of interest for shape–memory alloys [21], and in the food industry [22]. Experimental tests shall show whether 3D printing of rubber can be realized by a screw extrusion process. In the literature, various investigations as well as studies on the optimization of the parameters for rubber processing in screw extruders can be found [10,11,12]. The processing behavior is very complex, due to the mutual influence of the process parameters. Nevertheless, in 3D printing, parameters such as screw speed and plasticizing time are determined by the desired speed of the strand discharge. As a result, these parameters are no longer available as influencing factors as usual. Thus, previous approaches and methods of rubber extrusion cannot be transferred without further effort. For this purpose, feasibility tests are carried out and trials are performed to determine the ideal process parameters. Furthermore, the influences on the printing quality will be investigated. The consideration of the two further challenges for the supporting and stabilizing structures is not the subject of the current study. However, boundary conditions for further work can be derived from the results.

## 2. Materials and Methods

### 2.1. Materials

A natural rubber (NR) as well as the synthetic rubbers nitrile-butadiene rubber (NBR) and ethylene-propylene-diene rubber (EPDM) from the company Kraiburg (Waldkraiburg, Germany) [23], are available as test materials for 3D printing tests. For good processing and dispensing from the nozzle, the viscosity of the materials plays a particularly important role. Therefore, rubbers with a different viscosity (Mooney viscosity) than natural rubber were considered. On the one hand, a rubber with significantly lower viscosity (NBR) and on the other hand with significantly higher viscosity (EPDM) were selected. For all three materials, the Mooney and the vulcanization conditions are listed in the Table 1 according to the manufacturer’s specifications [23]. The materials are available in rubber sheets and are cut into strips of approx. 50 × 5 × 2 mm³ for use in the test rig.

### 2.2. Test Rig

The basis for the test rig for rubber compounds is an FDM printer. The printer had to be modified with a screw extruder, as is also usual for the processing of rubbers in injection molding. Among other things, the larger dimensions of a screw extruder and the higher weight resulted in the following requirements for the FDM printer:Open frame—for sufficient installation space and accessibilityFew moving axes on the extruder side—the influence of the additional weight of the screw extruder on the dynamics of the printer should be reduced as far as possibleDouble extruder—this provides the possibility to build supporting structures made of other materials (for further research work)Possibilities of software modification—possibilities of making specific adaptations to the program sequences for rubber processing.

The FDM printer M3-ID from MAKERGEAR [24], is chosen for the test rig. The printer meets all requirements. As a screw extruder, the pellet extruder is procured from the MAHOR XYZ shop [25]. The extruder, which is actually used for plastic pellets, is modified so that it can handle rubber strips (removal of fan, open structure of material feed). Figure 1 shows the modified 3D printer and the main parts of the screw extruder. After discharge from the nozzle, the rubber remains unvulcanized and therefore it does not attain a mechanically stable state like a thermoplastic. In contrast to conventional FDM printing, early and more extensive stabilization of the printed rubber on the platform is therefore necessary. This stabilization has not been implemented for the current study; therefore, a production of high or complex structures from rubber is not feasible in these experimental tests.

### 2.3. Experimental Tests

#### 2.3.1. Material Processing

Printing tests with the rubber materials were carried out on the test rig described above. The feed of the material into the screw extruder was evaluated. Furthermore, the continuous discharge of the strand at different temperatures was examined. For an adequate material transport in the extruder, a sufficient flowability must be provided. This can be achieved by heating the rubber material, which reduces the viscosity. However, the maximum temperature is limited, because above a certain temperature the vulcanization already starts in the screw extruder. This temperature value depends on the material that is used.

#### 2.3.2. Identification of Printing Parameters

The SIMPLIFY3D printing software is used to generate the G-code. For this purpose, the following parameters (illustrated in Figure 2a) in particular must be defined [26]:Extrusion multiplier (flow rate)Extrusion widthTraverse speedLayer height basic layerLayer heightNozzle temperatureTemperature printing bed.

Different nozzle diameters are used for the measurement of the strand diameter. It is known that due to the nozzle swell a strand expansion occurs after nozzle discharge [10,27]. The aim is to determine this change of the strand’s cross-section for the rubber material. While the printer head is stationary, the speed for the screw is set to a traverse speed (printing speed) of 40 mm/s and the resulting strand is used for the measurement (Figure 2b). In addition, the measurement is carried out for the speed of the screw with the traverse speeds (printing speeds) of 20 and 60 mm/s to evaluate the effects on the strand diameter. The measured value is taken as the value for the extrusion width. The nozzle temperature parameter is given by the results for material processing. The other parameters are initially set to default values as for a polylactide (PLA) print. These are adjusted iteratively until a good strand deposition is achieved. At the beginning, a glass plate is used as the printing platform. If a good adhesion of the rubber to the glass surface is not achieved, an adhesion promoter (3DLAC) or Kapton tape is available as an alternative surface.

#### 2.3.3. Print Quality

In 3D printing of thermoplastic materials, it is common to print test parts for setup or quality control. These usually have different two- or three-dimensional geometries [28]. For example, the achievable surface quality, the maximum possible overhangs without supporting structures, geometric dimensions and various three-dimensional shapes are evaluated. It can be assumed that overhangs or 3D geometries cannot be produced with the viscous rubbers, if no supporting structures are used. Therefore, the quality investigations in this paper focus on the dimensional accuracy and the examination of the strand laydown itself. The latter is performed as the first test.

The experiment is intended to investigate what layer height can be achieved without a supporting structure. It is unusual in 3D printing for layers to only be stacked on top of each other in one line. Thus, for this test double strand layers are stacked on top of each other according to Figure 3 with nozzle diameters of 0.4 and 0.8 mm. For this purpose, the print head deposits one strand on a length of 100 mm, makes a 180° curve, and then deposits a second strand on the same length in the opposite direction. Afterwards, another 180° curve is made, whereby the head now changes to the next layer height and again deposits such a double strand. On the one hand, the deposition of a strand after a change of direction can be evaluated and on the other hand the number of layers up to the visible change of the print quality can be counted.

The second test is performed to evaluate the quality of the dimensional accuracy of the printed shapes. For this reason, the two-dimensional test samples shown in Figure 4 were created, based on the usual test prints [28]. Test forms for FDM printing usually start in the range of 10 mm [28]. Accordingly, these dimensions are selected as the largest shapes in the test objects. The shapes become more filigree to examine and evaluate the compliance of the dimensional accuracy and the printing behavior for stronger changes of direction of the printer head. The quality of the inner contour is examined on Sample 1 (Figure 4a) and the outer contour on Sample 2 (Figure 4b). The minimum dimensions of the shapes for Sample 2 are restricted by the used nozzle diameter of 0.4 mm. For Sample 1 the minimum dimensions of the shapes are set to 0.1 mm. For the printing tests, the parameters listed in Table 2 are set as the basic values. The parameters are then varied individually according to the column “Parameter variation”. The shapes of the test samples with the best printing quality are measured in order to estimate the limits of the printing accuracy. A statistical measurement of several samples is not envisaged, as it can be assumed that the print quality depends mainly on the print parameters. Repeatability of the 3D prints depends on the accuracy of the axes. The manufacturer specifies a movement resolution of 1 µm per micro step, whereby a sufficient repeat accuracy is considered to be given.

## 3. Results and Discussion

### 3.1. Material Processing

During nozzle extraction of the materials NR and EPDM, blockages repeatedly occurred in the screw extruder. Up to a nozzle temperature of 90 °C, the viscosity was too high or the performance of the screw extruder drive was insufficient. Above 90 °C it was no longer possible to convey the two materials in the screw after a short time, which was a sign of the start of cross-linking in the screw extruder. Therefore, these two materials were excluded from further tests. In contrast, the NBR rubber showed good and stable strand discharge from the extruder at 70 °C. Consequently, all further listed tests were carried out with the rubber NBR. For further studies, it is recommended to install a stronger screw extruder drive in the test rig as well as to investigate the relationships between material data (Table 1) and processing behavior in 3D printing.

### 3.2. Identification of Printing Parameters

The measurement results for identification of the parameter extrusion width with a printing speed of 40 mm/s are listed in Table 3. The width was measured with a digital caliper with a measuring accuracy of ±0.02 mm and a resolution of 0.01 mm. Five measurements were carried out for each strand of length 500 mm. For a uniform measurement of the elastic material, the caliper was slid together stepwise until the strands could no longer move due to friction in the jaws. The minimum measured value, the maximum measured value, the average value and the average strand increase in comparison to the nozzle diameter are summarized in Table 3. It is shown that the maximum deviations within the strand width are less than ±0.02 mm from the mean value for each nozzle diameter. Thus, the strand width remains stable for every nozzle diameter. In comparison with a printing speed of 40 mm/s, the mean values of the strand diameters at a printing speed of 60 mm/s were less than 1% larger, and at the printing speed of 20 mm/s the mean values of the strand diameters were less than 1% smaller. The influence of these small deviations is considered negligible for the further tests. However, the strand increase in relation to all nozzles is within a range of 19% to 34% compared to the nozzle diameter. A dependence on the size of the nozzle diameters could not be recognized; this is explained by production-related differences. Therefore, it is recommended to measure the strand diameter when using a nozzle and then transfer the value to the software. The other parameters for a nozzle diameter of 0.4 mm, determined iteratively via the printing behavior, are listed in Table 4. For comparison, the default values for the PLA material, given by the software SIMPLIFY3D [26], are also presented in Table 4. Nearly all default values for PLA printing resulted in good strand laydown for the NBR rubber. The glass printing plate also proved to be a suitable printing platform. Only the extrusion multiplier and the height for the basic layer were adjusted. The extrusion multiplier had to be increased significantly from 0.9 to 5.0, as otherwise, the material quantity was not sufficient and strand interruptions occurred repeatedly. Reducing the height of the basic layer from 90% to 20% led to a significant improvement in adhesion to the glass surface.

### 3.3. Print Quality

In the first test, strands were stacked on top of each other with 3D printing. Five runs were performed with the nozzle having a diameter of 0.4 mm (layer height 0.2 mm) and five runs were performed with a nozzle having a diameter of 0.8 mm (layer height 0.4 mm). In each case, 7 to 8 double strands could be laid on top of each other until there was an optical shift of the layers (Figure 5). However, a major problem was found at the reversal point of the double strands. When changing the contour direction, considerable deviations had occurred in some cases from layer number 2 onwards, as shown in Figure 5 (side view).

In the second test, samples 1 and 2 were printed out of NBR and with the basic values from Table 2. The printed samples demonstrated good optical results, which are shown in Figure 6a,b. Furthermore, it can be seen that the nozzle drips from shape to shape. This can be avoided in the G-code through a stronger material retraction in the screw. Subsequently, the tests with parameter variations (Table 2) were realized. One parameter was changed and set back to the basic value before the next parameter was changed. The results for the different parameters are summarized in Table 5.

The parameter variation shows that for the current test rig the basic parameter values are already suitable for the test samples. For this reason, the dimensional accuracy of the shapes on the samples shown in the Figure 5 was measured with the UHX600 microscope at 20× magnification. The diameter of the circle *d* (Figure 7a), the width *w* of the rectangle (Figure 7b), and the height *h* as well as the upper angle *α* of the triangle (Figure 7c) were measured.

On the one hand, it can be seen in Figure 8; Figure 9 that for Sample 1 (inner contours) the shapes turned out to be smaller than the set point, which can later be corrected by the software. On the other hand, even smaller shapes could be produced than for test Sample 2. For Sample 2, the tolerances for the circle diameter are ±0.2 mm and for the rectangle width even ±0.1 mm. The tolerances are within a common range of ±0.5 mm for FDM printing with desktop printers [29]. All measured values for the circle diameter and the rectangle widths were below this tolerance.

Of course, the accuracy is also strongly dependent on the precision of the axes. For the triangle angles down to 20°, the tolerances are minimal and are below ±0.7° (Figure 10). However, the same problems can be observed with the small circles and triangles as with the double-strand layers (Figure 5). The smaller the diameter or angle of the shape becomes, the greater the change in direction for the nozzle. At an angle of 10°, the nozzle moves 170° and shows significant deviations in strand deposition compared to the other results. The increase in inaccuracies with increasing travel angle for the nozzle is also reflected in the measurement for the triangle heights (Figure 11). If the direction of the printer head changes rapidly or strongly, the strand can no longer be deposited well. This means that the rubber does not grip well on the lower layer and the material is slightly tightened. Especially during these printer head movements, the parameters in the software must be adjusted.

As a conclusion of the measurements, it can be stated that 3D printing of rubber can achieve a dimensional accuracy comparable to that of other printable materials using the FDM process. It becomes critical when circles are smaller than 4 mm and angles of 10° or smaller must be manufactured. In further investigations, it is necessary to check if a variation in traverse speed or the material discharge would provide better results, especially for such contours. An alternative nozzle geometry (e.g., rectangular nozzle opening) could also have a positive effect.

## 4. Conclusions

In this paper, fundamental investigations were carried out on the 3D printing of rubber. An FDM 3D printer was modified with a screw extruder and test prints were made. The deposition of strands on the table bed, the deposition of several strands on top of each other and the production of flat two-dimensional geometric shapes show the feasibility of 3D printing of rubber. Simultaneously, however, the enormous research potential for the development of a commercial process was revealed. There are two key challenges. Firstly, the stability of the printed object is influenced by the rheological material behavior and secondly, the implementation of the subsequent vulcanization process has not yet been solved.

The investigations have shown that in particular when the travel direction of the extruder is changed, inaccuracies in strand deposition occur. In order to solve this problem, further investigations into the correlation between strand deposition and the travel speed as well as the material flow rate at contours are necessary. Furthermore, it should be noted that due to the material behavior, structures could collapse even at low layer heights. One approach to solve this problem, for example, could be a material printed in parallel on all sides, which supports the overall structure from the first layer. At the same time, this medium can serve as a form stabilizer during a subsequent vulcanization process.

Based on the findings, the test rig should also be further adapted to the requirements for rubber printing. These future adaptations are, in particular, the reinforcement of the drive so that the force is sufficient to discharge different rubber material as well as an adaptation of the material feed, which is currently only designed for short rubber strips.

## Figures and Tables

**Figure 1 polymers-12-02266-f001:**
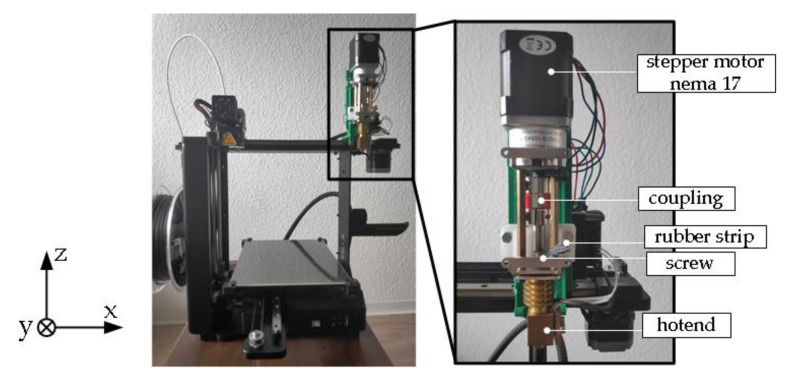
The MAKERGEAR M3-ID desktop 3D printer modified with a screw extruder.

**Figure 2 polymers-12-02266-f002:**
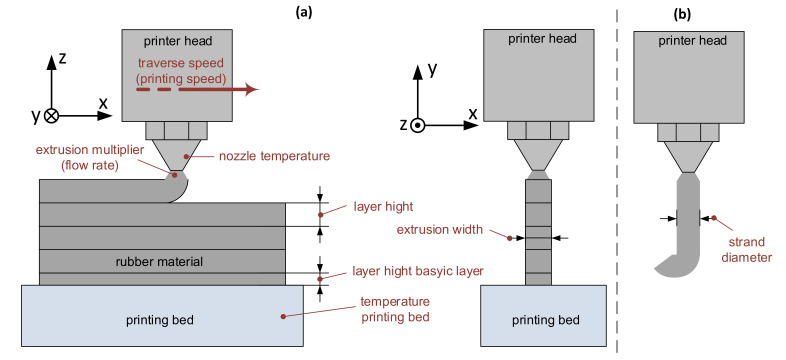
(**a**) Visualization of the parameters to be set for 3D printing and (**b**) discharge of the rubber material out of the nozzle, for measuring the strand diameter.

**Figure 3 polymers-12-02266-f003:**
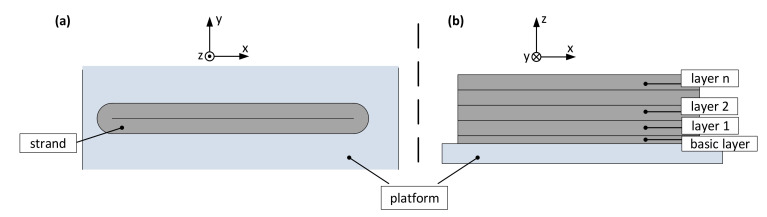
Test procedure—stacking double strands with 3D printing: (**a**) the top view and (**b**) the side view.

**Figure 4 polymers-12-02266-f004:**
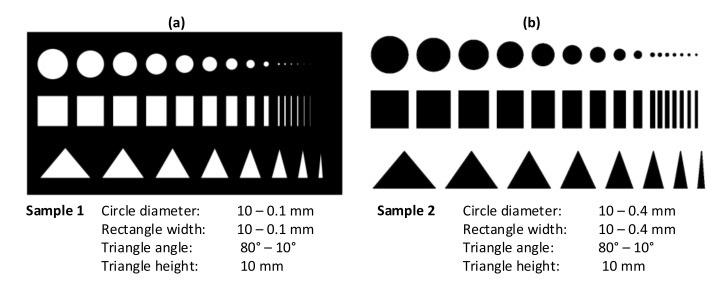
The models designed for 3D rubber printing based on the usual test prints [28]: (**a**) Sample 1 with geometric shapes as the inner contour and (**b**) Sample 2 with geometric shapes as the outer contour (the minimum dimensions are limited by the nozzle diameter of 0.4 mm).

**Figure 5 polymers-12-02266-f005:**
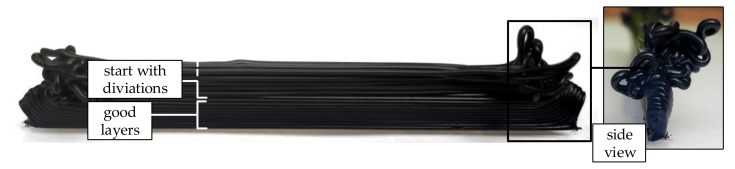
The printing result for stacking double strands. Seven to eight layers were laid well on top of each other until deviations occurred. The trigger of the deviations is probably caused by the reversal point, as the side view illustrates.

**Figure 6 polymers-12-02266-f006:**
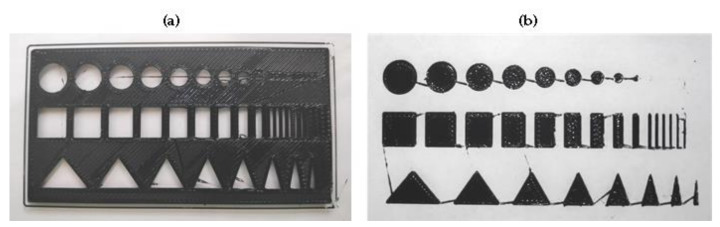
Examples of printing results with the nozzle diameter of 0.4 mm. The prints show a good formation of the geometric forms, but also a nozzle dripping: (**a**) Sample 1 and (**b**) Sample 2.

**Figure 7 polymers-12-02266-f007:**
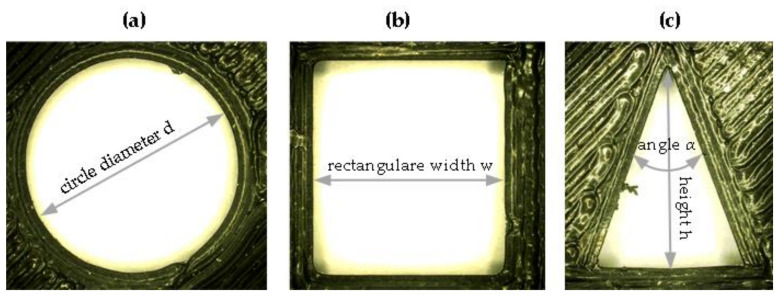
Measurement parameters of the geometric shapes: (**a**) the circle diameter *d*, (**b**) the rectangle width *w* and (**c**) the triangle height *h* and angle *α*.

**Figure 8 polymers-12-02266-f008:**
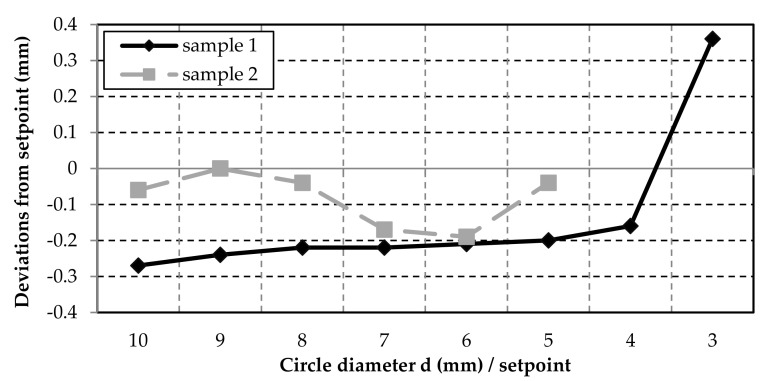
The deviations from the set point of the inner (Sample 1) and outer (Sample 2) circles.

**Figure 9 polymers-12-02266-f009:**
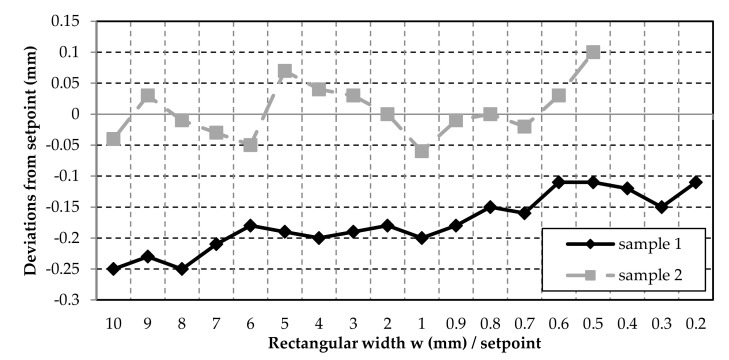
The deviations from the set point of the inner (Sample 1) and outer (sample 2) rectangle width.

**Figure 10 polymers-12-02266-f010:**
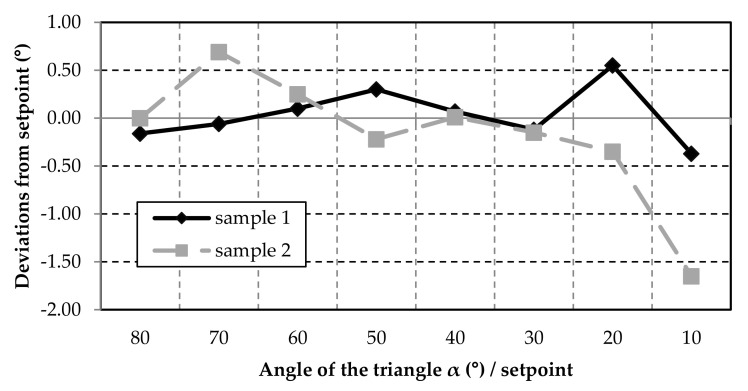
The deviations from the set point of the triangle angles of Sample 1 and Sample 2.

**Figure 11 polymers-12-02266-f011:**
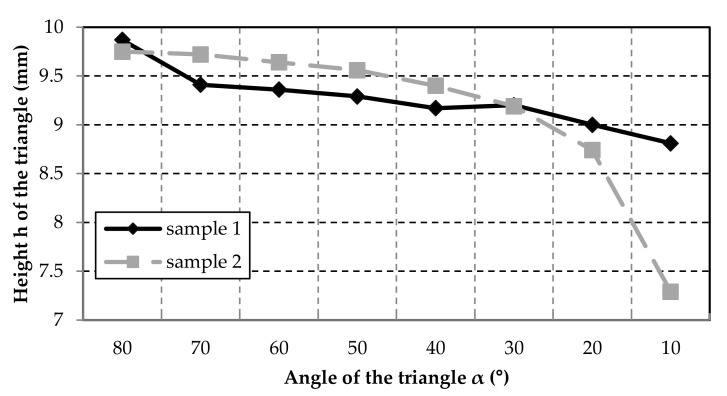
The height of the triangles of Sample 1 and Sample 2 at different angles. The set point of the height for all triangles is 10 mm.

**Table 1 polymers-12-02266-t001:** The Mooney viscosity of the available rubbers (according to the manufacturer [23]).

	Natural Rubber (NR)	Ethylene-Propylene-Diene Rubber (EPDM)	Nitrile-Butadiene Rubber (NBR)
Mooney viscosity (ML1 + 4; 100 °C)	57	112	37
Vulcanization Conditions Dumbbell specimen S2 (10 min)	160 °C	170 °C	170 °C

**Table 2 polymers-12-02266-t002:** The basic values and parameter variations for the printing tests.

Parameters	Basic Value	Parameter Variation
Traverse speed	40 mm/s	20 mm/s, 60 mm/s
Nozzle diameter	0.4 mm	0.8 mm
Number of layers	Basic layer + 1 layer	Basic layer + 5 layers

**Table 3 polymers-12-02266-t003:** The measurement results for the strand diameter after the nozzle outlet (with a printing speed 40 mm/s).

Nozzle Diameter.	Measured Diameter		Mean Value	Increase to Nozzle Diameter
	Min	Max		
(mm)	(mm)	(mm)	(mm)	(%)
0.25	0.33	0.34	0.334	34
0.3	0.36	0.37	0.364	21
0.4	0.52	0.54	0.524	31
0.5	0.64	0.67	0.654	31
0.6	0.73	0.77	0.746	24
0.8	0.94	0.95	0.948	19
1	1.2	1.22	1.212	21
1.5	1.81	1.82	1.816	21

**Table 4 polymers-12-02266-t004:** Printing parameters for NBR compared with PLA.

	NBR	Polylactide (PLA) (Default Values of the Software [26])
Extrusion multiplier	5.0	0.9
Extrusion width	Results from Table 3	Nozzle diameter + 20%
Traverse speed	40 mm/s	40 mm/s
Layer height basic layer	20% Nozzle diameter	90% Nozzle diameter
Layer height	50% Nozzle diameter	50% Nozzle diameter
Temperature pressure bed	50 °C	60 °C
Printing surface	Glass	Glass

**Table 5 polymers-12-02266-t005:** The parameter variation and their effect on the 3D printing.

Parameters	Effects on Printing
Decreased traverse speed 20 mm/s	The lower traverse speed caused worse results compared to the base value (40 mm/s). A good basic layer could be produced. However, in the subsequent layer, deviations in the contours occurred more frequently when the contour direction was changed. This could not be noticed with the base value. The reason for this could be that a minimum discharge speed from the nozzle is required for a good layer-on-layer adhesion with rubber. Therefore, if a strand is placed on another strand with less pressure, the adhesion will be reduced and the strands will tilt or shift.
Increased traverse speed 60 mm/s	The higher traverse speed caused a worse result than the base value (40 mm/s). A higher traverse speed also leads to a higher material transport in the screw. However, this leads to a steady blocking of the screw and thus to reduced material discharge and strand interruptions. The reason for this is assumed to be that the material feed into the screw could not be heated up fast enough in the extruder to the nozzle outlet. As a result, the viscosity is not lowered far enough for further transport in the screw. A solution is to extend the screw length and thus the temperature control zone. It would also be possible to increase the drive power of the screw extruder or to heat the rubber before feeding it into the screw.
Increased extrusion width 0.8 mm	For the nozzle diameter of 0.8 mm, both the basic layer height of 0.2 mm and the layer height of 0.4 mm (the half nozzle diameter) were tested. As expected, the smaller geometric shapes were not manufactured as precisely. Furthermore, with the layer height of 0.4 mm the same problem occurred as observed with increasing the traversing speed. The screw extruder was not able to process the increased rubber quantity.
Basic layer + 5 layers	The printing of the basic layer and the first layer showed a visually good result. From layer 2 on, first deviations occurred, which increased significantly from layer to layer: The smaller the dimensions of the geometric form, the greater the deviations. As an example, Figure 5 shows the manufactured shapes. Further research should investigate whether modified nozzle geometries produce better results. In general, the viscous material has to be supplemented by another, all-round supporting material to produce higher structures.

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
