# Peer review of "Basic Research for Additive Manufacturing of Rubber"

_polymers, 2020, doi:10.3390/polym12102266_

Round 1
Reviewer 1 Report
Dear Authors,
The manuscript is an excellent contribution to the understanding of the rubber processing by 3D printing. However, I have a little concern, although three rubbers were mentioned in materials section (NR, NBR and EPDM); apparently, the compositions of sample 1 and sample 2 were not included. Could you provide further information of the sample 1 and 2 composition?
Best wishes
Author Response
Dear Reviewer,
thank you for taking the time to review our work and for the good criticism.
"Could you provide further information of the sample 1 and 2 composition?"
The problem with the extrusion based setup was that only the material NBR was printable, which is described in section 3.1. For a better understanding, line 275 was added to show that all samples were made with NBR.
Sincerely,
Marek Schmidt
Reviewer 2 Report
This paper presented an interesting investigation of 3D printing of rubber materials using extrusion based 3D printing. The authors first reviewed the state of the art in this field. Then, the materials, 3D printing setup, 3D printing process and characterization were addressed. The presented results were able to support the scientific discoveries. However, the reviewer believes the paper can be improved by clarifying certain details. Please see the detailed comments below.
1. The introduction can be improved by discussing what type of extrusion based 3D printing on what materials have been developed. The authors are not the first group to employ extrusion based 3D printing. It is better to refer to some of other people's work and also discuss the pros and cons of current technologies.
2. Please clarify the parameters reported in Table 3. The reviewer believes the strand diameter is a function of extrusion rate and travel speed of the printer head. Please provides the two speeds for Table 3 and discussion the variation of strand diameter when the two speeds change.
Author Response
Dear Reviewer,
thank you for taking the time to review our work and for the good criticism.
“1. The introduction can be improved by discussing what type of extrusion based 3D printing on what materials have been developed. The authors are not the first group to employ extrusion based 3D printing. It is better to refer to some of other people's work and also discuss the pros and cons of current technologies.”
Thank you for pointing this out. Other works on extrusion-based 3D printing were added in line 96-102 (with track changes). These lines also contain the advantage of the process that rubber strips can be used as feed material. The naming of disadvantages cannot be generalized, because it depends especially on the used material and other individual technical factors. For rubber processing itself, the aim of the paper was to identify the disadvantages.
"2. Please clarify the parameters reported in Table 3. The reviewer believes the strand diameter is a function of extrusion rate and travel speed of the printer head. Please provides the two speeds for Table 3 and discussion the variation of strand diameter when the two speeds change."
Thanks for the suggestion to check this fact. In the experiments, the strand was taken directly from the nozzle for measurement. Thus, it was independent of the travel speed. The speed of the screw, which ultimately controls the material feed, was now additionally varied (via the printing speed in the software). Slight deviations (<1%) from the previous measured values were recorded. This is considered negligible and is included in paper in the lines 176 – 180 and 257 - 260.
Sincerely,
Marek Schmidt
Reviewer 3 Report
In this paper, the authors tried to propose a 3D printing approach for rubber printing. But I didn’t see the innovation of this work. Screw-driven 3D printers have been developed and used for many years. What were the unique contributions of this paper? In addition, there were some technical issues to address before this paper can be published as follows:
- The limitations of current 3D printing approaches in printing vulcanizable rubber were not well-presented.
- Why were NR, NBR, and EPDM selected as the build materials for printing? Any reasons?
- In section 2.3.2, one schematic should be added to show the printing parameters, otherwise, it’s hard to understand the meaning of each parameter.
- How to define print quality in this paper?
- What’s the meaning of “double strand”? A schematic was needed to make it clear. Why were double strands printed, not stacking single strands layer-by-layer? The authors wanted to investigate the interface between two horizontally adjacent strands?
- Figs. 7-10 were used to quantitatively present the printing fidelity of the samples as shown in Fig. 5. Why were these dimensions selected? How many samples were tested? How about relative errors or standard deviations?
- Any discussions and explanations for the changes of dimensions in Figs. 7-10? Such explanations can help understand the effectiveness of this rubber printing method.
Author Response
Dear Reviewer,
thank you for taking the time to review our work.
To the opening words: "What were the unique contributions of this paper?"
The aim of the paper was not to present the approach of extrusion-based 3D printing, but to use it as a means to an end. In order to clarify this, the lines 99 – 103 (line numbers always with track changes) show that there is already a significant amount of research work in this area.
The focus was on the investigation of rubber processing and the behavior during and after 3D printing to enable first steps towards opening up the material for this technology. The authors are not aware of any comparable work in this field.
- The limitations of current 3D printing approaches in printing vulcanizable rubber were not well-presented.
Lines 82 to 88 were added to clarify the limitations of printing vulcanizable rubber.
- Why were NR, NBR, and EPDM selected as the build materials for printing? Any reasons?
The aim was to consider the effects of different viscosities on processing in 3D printing of rubber. Unfortunately, only NBR was printable with the test setup. Additions were made in lines 120-122.
- In section 2.3.2, one schematic should be added to show the printing parameters, otherwise, it’s hard to understand the meaning of each parameter.
A schematic for the parameters was added in line 171 (Figure 2).
- How to define print quality in this paper?
Thank you for pointing this out. For this purpose, section 2.3.3 has been extensively revised and supplemented with the definition of the print quality in this paper.
- What’s the meaning of “double strand”? A schematic was needed to make it clear. Why were double strands printed, not stacking single strands layer-by-layer? The authors wanted to investigate the interface between two horizontally adjacent strands?
This point was also described in more detail with the revision of section 2.3.3 (lines 194 - 202). Figure 3 is part of the explanation.
- Figs. 7-10 were used to quantitatively present the printing fidelity of the samples as shown in Fig. 5. Why were these dimensions selected? How many samples were tested? How about relative errors or standard deviations?
Simple geometries were chosen for the test objects. The aim was to investigate how exactly the dimensional accuracy can be maintained with decreasing dimensions. The dimensions of the largest shapes are based on standard test print objects. In order to clarify this even more, lines 216 - 219 were added.
After determination of optimal printing parameters, 3 samples were produced. One was measured in detail (results in paper). For the other two samples, especially the dimensions with the larger deviations were also measured. The measured values were the same with negligible deviations. As now added in line 225 - 229, it can be assumed that the reproducibility with the same printing parameters depends on the accuracy of the axes. The manufacturer specifies a high accuracy for the axes, therefore errors or standard deviations are not considered in this context.
- Any discussions and explanations for the changes of dimensions in Figs. 7-10? Such explanations can help understand the effectiveness of this rubber printing method.
Deviations from the dimensions for the larger shapes are in a common range for 3D printing as described in line 307-312. The reason for the deviations in the smaller shapes is the worse strand deposition, which is explained by the larger changes of direction of the printer head. Supplements to this have been inserted in lines 319 - 322.
Sincerely,
Marek Schmidt
Round 2
Reviewer 3 Report
I was still not convinced by the revised manuscript regarding the significance and unique contribution. The authors summarized the technical challenges in rubber 3D printing. I agreed. But what's the proposed solution? Material modification or new 3D printing technique? From Introduction, the only contribution seems to be using screw extruder to replace the conventional extruder in FDM. If that's the case, the authors need to further explain why screw extruder is the only feasible solution for rubber printing. If the authors just used an existing technique to print commercially available build materials, what're the novelty and significance?